# Elaborative Simplification as Implicit Questions Under Discussion

**Yating Wu**[*1]   **William Sheffield**[*2]   **Kyle Mahowald**[2]   **Junyi Jessy Li**[2]
[1]Electrical and Computer Engineering, [2]Linguistics
The University of Texas at Austin
{yating.wu, sheffieldw, mahowald, jessy}@utexas.edu

## Abstract

Automated text simplification, a technique useful for making text more accessible to people such as children and emergent bilinguals, is often thought of as a monolingual translation task from complex to simplified text. This view fails to account for *elaborative simplification*, where new information is added into the simplified text. This paper proposes to view elaborative simplification through the lens of the Question Under Discussion (QUD) framework, providing a robust way to investigate *what writers elaborate upon*, *how they elaborate*, and *how elaborations fit into the discourse context* by viewing elaborations as explicit answers to implicit questions. We introduce ELABQUD, consisting of 1.3K elaborations accompanied with implicit QUDs, to study these phenomena. We show that explicitly modeling QUD (via question generation) not only provides essential understanding of elaborative simplification and how the elaborations connect with the rest of the discourse, but also substantially improves the quality of elaboration generation.

## 1 Introduction

Text simplification systems aim to lower the barrier of reading for a wider, more inclusive audience, for instance, children (De Belder and Moens, 2010), emergent bilinguals (Taylor et al., 2022), and individuals with language impairments (Carroll et al., 1998; Rello et al., 2013). While there has been abundant research in automatic text simplification (Siddharthan, 2014), recent data-driven efforts have focused on re-writing a sentence or passage into simpler language while preserving its meaning, often as a monolingual translation task using encoder-decoder models (Alva-Manchego et al., 2020; Sun et al., 2021; Devaraj et al., 2021) or editing models (Dong et al., 2019; Agrawal and Carpuat, 2022).

---

* Yating and William contributed equally.

> **Simplified.** Those factories are gone now. New companies have come that need skilled workers with more education. New Haven youth want those jobs, but they do not have the education or the skills.
>
> *Implicit QUD: Why don't people acquire the necessary skills?*
>
> Many do not have the money to get the training they need.
>
> That is where New Haven Promise comes in. It will make a difference by paying for college. New Haven Promise is no one-way street.

**Original.** Factories have closed and their low-skill manufacturing jobs are long gone. The new companies in town require workers with a college degree or advanced training. […] The New Haven Promise is part of a bigger plan to improve the city's economy.

Figure 1: An example of elaborative simplification, taken from Srikanth and Li (2021). Both simplified and original snippets are shown; elaboration added to the simplified version is shaded in blue. "[...]" in the original text refers to content deleted in the simplified version. **This work** focuses on already identified elaborations in the simplified text, and introduces implicit questions under discussion ("implicit QUD", yellow box) to characterize and help generate the elaborations.

This work instead focuses on *elaborative simplification* (Srikanth and Li, 2021), i.e., explaining or elaborating difficult concepts or content during the simplification process, as illustrated in Figure 1. Although elaborations would add to the amount of content a reader needs to process, psycholinguistic studies have established the benefit of elaborative modifications for L2 reading comprehension (Parker and Chaudron, 1987; Yano et al., 1994). However, deriving elaborative simplification is challenging: existing simplification models—because they are trained as end-to-end translation models—do not actively generate elaborations and, when they do, tend to hallucinate (Devaraj et al., 2021; Srikanth and Li, 2021). Thus to make progress, we argue that explicit analysis and supervision is necessary. There has been little

work understanding *what people choose to elaborate, how they elaborate, and how the elaboration fits into the discourse context*. Understanding these dimensions is crucial for developing better systems by giving us a framework for analyzing elaborations.

We propose a simple but powerful way of thinking about elaborations: as answers to implicit questions. Consider Figure 1: the editor inserted "*Many do not have the money to get the training they need*" as an explanation for the preceding sentence "*they do not have the education or the skills*". This elaboration did not exist in the original (unsimplified) document, and it can be thought of as answering the implicit question "*Why don't people acquire the necessary skills?*".

This approach has a long history in the Question Under Discussion (QUD) linguistics framework (Von Stutterheim and Klein, 1989; Van Kuppevelt, 1995; Roberts, 2012; Benz and Jasinskaja, 2017; Ko et al., 2023); the QUD framework views each sentence as the answer to an implicit or explicit question from prior context. Thus, our model for elaborative simplification is that, while simplifying text, editors implicitly ask questions, especially when difficult concepts are encountered. Elaborative simplifications are (explicit) answers to these (implicit) questions.

With this view, we formulate elaborative simplification as a two-step process: question generation and question answering. The question generation step models "what is elaborated?" by means of *recovering* the implicit QUDs, which will guide the question answering model for the generation of the actual elaboration.

To support this, we present ELABQUD, a novel corpus of implicit QUDs that are answered by the 1299 elaborations collected by Srikanth and Li (2021). In addition, ELABQUD also contains a finer-grained layer of annotation specifying which concepts the elaboration was about in the earlier context of the same document, which we call the *targets* of elaboration. We find authors elaborate both about entities and events, and that elaborated concepts tend to be composed of less frequent words. We also analyze the types of questions to determine how authors elaborate, and find that elaborations often involve causal reasoning and explanations of entities.

Using ELABQUD, we first train and evaluate question generation models attempting to automat-

ically generate the QUDs. We train these models using two QUD corpora, then fine-tune on ELABQUD: one setting where the model is exposed to the elaboration (Ko et al., 2022), and one where the model is not (Ko et al., 2020) following the expectation-driven model of QUD (Kehler and Rohde, 2017). The latter setting mimics the realistic scenario where the answer (i.e., the actual elaboration that we aim to generate) is *not* known prior to asking the questions. We show that expectation-driven questions, although often plausible and valid, tend to deviate more often from the exact direction of the annotated QUDs.

Next, we plug in the generated questions as prompts for a GPT-3 model (Brown et al., 2020) to derive elaborations in a zero-shot manner. We show that compared with no prompt or generic prompts, QUD-driven elaborations are of substantially higher quality and are typically more elaboration-like.

We release ELABQUD and code at `https://github.com/sheffwb/elabQUD` (copyright issues discussed in Appendix C).

## 2 Background and Related Work

**Elaborative Simplification** Earlier work related to elaborative simplification mostly focused on a specific type of elaboration, namely retrieving definitions in lexical simplification (Damay et al., 2006; Kandula et al., 2010; Eom et al., 2012). More recently, Srikanth and Li (2021) gathered a general dataset for elaborative simplification, all of which were derived from the Newsela dataset (Xu et al., 2015), a corpus of professionally simplified news articles. The elaborations were obtained by first finding sentences in the simplified version of a document that failed to align to the original version. These candidates were then manually filtered via crowdsourcing to check whether they appeared in a context window in the original version.

Srikanth and Li (2021) found that only some of the inserted elaborations were definitions; many were contextually dependent explanations and clarifications (e.g., Figure 1). In a few cases, editors would choose to add additional facts related to an event. This rules out definition retrieval as a full solution to the elaboration generation task. Additionally, Srikanth and Li (2021) showed that vanilla use of an auto-regressive language model could generate ersatz "elaborations" that deviate from the document context, hallucinate, and/or do not

actually explain the content.

**Questions Under Discussion**  The QUD framework is a way to reason through discourse by viewing it as continuously posing and answering questions (Von Stutterheim and Klein, 1989; Van Kuppevelt, 1995; Roberts, 2012; Benz and Jasinskaja, 2017). In dialogues, participants actively resolve the current QUDs; however in monologues, the QUDs are implicit. Thus in this work we *recover* the implicit QUD that was triggered in context prior to the elaboration that answers it. Recent work has begun to explore the annotation (De Kuthy et al., 2018; Hesse et al., 2020; Westera et al., 2020; Ko et al., 2020, 2022) and automatic generation (Ko et al., 2023) of QUD structures. Our data collection process aligns with Ko et al. (2022)'s annotation paradigm for QUD recovery, wherein each sentence of a news document is considered an answer to a QUD from prior context. In our case, each elaboration is the answer to a QUD that comes up when the need for more explanation arises.

Despite decades of rich linguistic research on QUD, large-scale, task-oriented application of this framework is still in its infancy, with very recent efforts studying question generation (Ko et al., 2020) and answering (Ko et al., 2022), conditional generation (Narayan et al., 2023), and decontextualization (Newman et al., 2023). The goal of this work is to lay a foundation connecting elaborations with QUD: given a marked elaboration, using QUDs to characterize what and how the elaboration should be generated. Although this work does not address *when* an elaboration should be added (which we leave for future work), the QUD framework provides a natural, interactive, and personalized way to think about elaborations: the QUD will be explicitly provided by the reader when they think more explanation is needed.

## 3  Implicit QUDs: what questions do elaborations answer?

For a simplified document with sentences $D = \{S_1, ..., S_{i-1}, S_i, S_{i+1}, ...\}$ where sentence $i$ is an elaboration (i.e., $E = S_i$), we aim to recover the implicit question under discussion $Q$ such that $E$ answers $Q$. We further define the target $T$ of the elaboration, i.e., $E$ elaborates or explains $T$. The sentence that contains $T$ is called the *anchor sentence* of $Q$, and can be taken to mean that $Q$ arose from that anchor (Ko et al., 2022).

❶ Read elaboration (highlighted) and context

> **1** The Giron brothers left Honduras in February 2013. […] **4** It took the group 11 days to travel from Honduras to the United States. **5** They traveled all the way north through Mexico until they reached the Rio Grande River.
> The river divides the United States from Mexico.
> At the border, the guide put the Girons and other teens on a small raft. He paddled them across the river to Texas. Once they were in the United States, their guide left the teens by themselves.

❷ Formulate question

*Why is it important for them to reach the Rio Grande?*

❸ What concept, and in which sentence, does the elaboration explain?

*Sentence 5, "the Rio Grande River"*

❹ Is this elaboration an organizational sentence?

■ Yes  ☑ No

Figure 2: Annotation procedure of ELABQUD.

This section presents ELABQUD and its annotation. ELABQUD contains annotated implicit QUDs for all 1,299 elaborations in Srikanth and Li (2021), along with their anchors and targets.

### 3.1  Annotation task

Our annotation process is depicted in Figure 2. We adapt Ko et al. (2022)'s annotation paradigm for less cognitive load since we focus on one elaboration at a time, and we introduce task-specific modifications. Specifically, for a given elaboration $E$, annotators were provided with a context window of five sentences preceding $E$, $E$ itself, and the three sentences succeeding $E$.[1] We show five prior sentences as in Srikanth and Li (2021), who found that this is usually sufficient and effective to establish common ground. The three succeeding sentences were shown to provide a more rounded picture of the document, although this information is not necessary for the annotations. For ease of reading, the elaborations were highlighted in yellow.

Next, the annotators were asked to create questions which were (a) plausibly generated by considering only the context, and (b) for which the elaboration provides an answer. To better simulate the real elaboration simplification process where $E$ is unknown when the question is asked, we ask annotators to avoid including content specific to $E$

---

[1]At the edge of documents, annotators were provided as many lines as possible.

in the questions.

We then ask the annotators to identify the target $T$ that $E$ elaborates. After the first round of annotations both by the authors and by crowdsourced workers, we found that, in most cases, both the anchor sentence and $T$ were in the sentence immediately preceding the elaboration (i.e., $T \in S_{i-1}$ when $E = S_i$), and that with multiple analyses $S_{i-1}$ usually provided the most straightforward $T$. Thus, we also highlighted $S_{i-1}$ in the interface. However, when asking annotators to provide $T$, we did not prime them further to $S_{i-1}$, and allowed them to highlight as $T$ any subsequence in the prior context that they deem plausible.

Finally, we noticed that some sentences are *organizational*: they are added to provide discourse cues that describe the way the next few sentences are organized, e.g., the elaboration text **E** in the example below. We included an additional question to mark these.

> (1) Investigators say Kellogg tried to copy the watermark.
> **E**: Here's how they say he did it.
> First he printed the front side of the money on one piece of paper. Next, [...]

**Annotators** The primary annotation task had two stages. The first stage involves three expert annotators at our institution who each annotated the same 30 elaborations. From these, we identified a representative set of six elaborations for which all annotators agreed on the target $T$ and asked semantically equivalent questions to form a worker qualification dataset. Their feedback was also used to enhance instructions and guide minor improvements to the annotation interface.

The full dataset was then collected via crowdsourcing using Amazon Mechanical Turk. Annotators that had previously worked with our institution on other complex document comprehension tasks were asked to annotate the six qualification elaborations as a qualification task. Responses were manually inspected, and those that matched the expert target annotations and gave highly similar or reasonable alternative questions were qualified. In total, 8 workers were approved. They were paid at a rate of over $10 per hour. Each elaboration was annotated by 2 annotators (with a subset of 280 annotated by 3 annotators); in total, we collected 2,878 questions across the 1,299 elaborations in Srikanth and Li (2021). The collected questions had an average length of 8.80 tokens (std.dev 3.25).

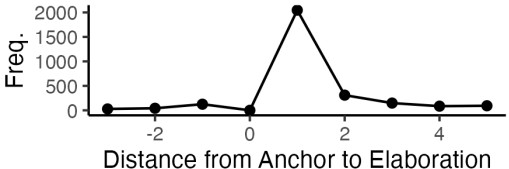

Figure 3: Distribution of distance from anchor sentence to elaboration.

## 3.2 Analysis

**Are questions similar for the same elaboration?** We report BERTscore (Zhang et al., 2019) between each pair of questions. We include both raw and rescaled[2] values. Annotator questions have a BERTscore F1 of 0.922 (rescaled 0.538). Compared to randomly-paired questions from the same article (F1 0.879; rescaled 0.281), these values indicate high similarity between questions from different annotators for the same elaboration when compared to random question pairings.

For the anchor sentence, we measure agreement based on the distance from it to the elaboration, meaning a distance of 3 indicates the anchor sentence occurs 3 lines before the elaboration, while a distance of -1 indicates the anchor sentence occurs in the line after the elaboration. The distribution of distances is provided in Figure 3; most anchor sentences immediately precede $E$. We observe a Fleiss' kappa (Fleiss, 1971; Randolph, 2005) of 0.6083 (percentage agreement 69.9%), indicating substantial agreement (Artstein and Poesio, 2008). Additionally, the selected targets overlap 62.4% of the time, reflecting that annotators agree on what is being elaborated most of the time.

**What is elaborated?** Although the average target is 4.54 tokens long, there is considerable variation (standard deviation of 3.06). Nouns are the most frequent part of speech in the targets (7452), specifically plural nouns (1589) and proper nouns (1449) out of a total number of 13153 tokens. These are often the targets of definitions, or something along those lines. For instance, the first example in Table 1 has an entity target that explains more about the entity without being an explicit definition. Moreover, we surmise a significant subset of elaborations focus on entities because 31.4% of all targets contain proper nouns.

---

[2]Rescaling is provided by the original authors to improve model interpretability as the original scores are often close "potentially because of the learned geometry of contextual embeddings" (Zhang et al., 2019).

| Question Type Definition | Example from ELABQUD |
|---|---|
| **Concept (34%)**: Asking for a definition of an event or a concept. | Anderson became interested in people like Landa when she noticed something strange about a call center near her house. [**Q**: *What do call centers do?*] **E**: Workers at call centers help people over the phone. |
| **Example (16.2%)**: Asking for example(s) or instance(s) of an event or a concept. | The government is split into two parties that often have different political beliefs. [**Q**: *What is an example of one of these parties?*] **E**: One party is the Democrats. |
| **Consequence (13.9%)**: Asking for the consequences or results of an event. | The tightropes that Wallenda walks across go between buildings, hundreds of feet above the ground. [**Q**: *What if he falls?*] **E**: There are no nets to catch him if he falls. |
| **Cause (12%)**: Asking for the cause or reason for an event or a concept. | But not many countries support Obama's plan to fire missiles at Syria. [**Q**: *Why are they being unsupportive?*] **E**: Some are worried about getting into another war in the area without knowing the facts. |
| **Procedural (8.1%)**: Asking for the procedures, tools, or methods by which a certain outcome is achieved. | The drone safely flew above the Atlantic Ocean and landed on an aircraft carrier called the George H.W. [**Q**: *How did the drone navigate its way to aircraft carrier?*] **E**: It was given special directions from satellites above the earth. |

Table 1: Top question types, their definitions from Cao and Wang (2021), and examples.

While many targets comprise noun phrases, 48.99% of targets include a verb, indicating that writers elaborate on events as well as entities. Take, for instance, the organization example (1) stated earlier. In this example, the target ***copy the watermark*** contains a verb and the elaboration focuses on the event of copying the watermark rather than Kellogg or the watermark itself.

We also found that authors usually elaborate on less frequent words. We measured this using log frequency per million words from the SUBTLEX-US (Brysbaert et al. (2012), 2015 release) corpus. The average log frequency values (per million words) for targets is 1.72, significantly lower than the document average of 2.46 (by an independent-samples t-test, $t = -34.5, p < .00001$).

**What types of questions are asked?** To examine the types of questions, we classify the questions collected using the taxonomy and model from Cao and Wang (2021). In Table 1, we show the top 5 question types in ELABQUD along with examples. The implicit QUDs reveal that in most cases, the elaboration is explaining a concept (34%), providing explicit causal reasoning by describing the cause (12%) or consequences (13.9%) of an event, providing an example (16.2%), or describing a complex process (8.1%). Other question types (e.g., verifying the truthfulness of a concept, comparison among multiple events, or asking about the extent of an event) are rare, indicating that the communicative goal of an elaboration in the Newsela dataset is to provide an explanation when reasoning is deemed difficult for children.

We additionally present an analysis connecting elaborations with expert-annotated discourse relations on a small set of 40 examples. We observe intuitive correspondences between discourse relations and question types, detailed in Appendix A.

## 4 Question generation

With the QUD framework, elaborative simplification is a two-step process:

**(1)** given context $C = S_1, S_2, ..., S_{i-1}$ prior to the elaboration $E = S_i$, generate a question $Q$ to recover the implicit QUD by modeling $P(\mathbf{q}|C)$.

**(2)** Given $C$ and $Q$, generate elaboration $E$ by modeling $P(\mathbf{e}|C, Q)$.

This section experiments with question generation (QG) models for **step (1)**. We explore three different settings varying how explicitly the model sees the elaboration target $T$ and the anchor sentence, and establishing an upper bound where the model is exposed to the gold "answer" $E$.

### 4.1 Models

**Oracle setup: QG model sees $E$.** Knowing the answer would inform a QG model what questions to ask. Although our target model will not see the answer (as it is generating a question used in-turn to generate the answer/elaboration), we can use such a QG model as a silver upper-bound on QUD generation. Here we repurpose the DCQA dataset (Ko et al., 2022) for question generation. DCQA consists of 22K questions from ~600 news articles; these questions are implicit QUDs elicited from annotators where they considered each sentence of these articles as an answer to a question. Each question is associated with an anchor sentence that triggers the question (the anchor sentence contains

the target $T$ but DCQA does not annotate $T$) and an answer sentence. In our case, we include all sentences prior to $E$, along with $E$, to see how they help compose questions about $E$.

We first fine-tune GPT2-medium (Radford et al., 2019) on DCQA with the input consisting of prior context $C$, the anchor sentence, the answer sentence $E$, and annotated question $Q$ with special delimiters separating each part. We call this model **DCQA-base**. We then fine-tune DCQA-base on ELABQUD, which we call **DCQA-ft**. We refer readers to Table 7 (Appendix) for a listed view of model inputs to *all* systems.

**Practical system: QG model does not see $E$.** Realistically, since $E$ is what we eventually want to generate, the QG model cannot not be exposed to it. This paradigm fits with the *expectation-driven* approach to QUD (Kehler and Rohde, 2017), where the questions are more curiosity-driven and are asked without seeing upcoming context.

Thus we train our QG model using the INQUISITIVE dataset (Ko et al., 2020), the largest question generation dataset annotated in line with an expectation-driven approach. INQUISITIVE consists of ∼19K questions elicited as the reader sees the first 5 sentences of a news article one by one (without access to the document as a whole). INQUISITIVE also includes target annotation in the anchor sentence where the question was raised; this allows us to experiment with models that explicitly predicts the target $T$.

Specifically, our model **INQ-GoldT-base** is from Ko et al. (2020), a GPT-2 medium model fine-tuned on INQUISITIVE. The input to this model includes all sentences prior to the anchor sentence, the anchor sentence itself including the *gold* target span $T$ marked, and the annotated question $Q$ with special delimiters separating each part.[3] We then fine-tune this model on ELABQUD, which we call **INQ-GoldT-ft**.

Our second INQUISITIVE model, **INQ-PredT**, involves a pipeline approach that first predicts $T$. We following the same setting as Ko et al. (2020): we train a `distill-bert-uncased` model with a modified SQuAD-QA format.

---

[3] We do not predict the anchor sentence; at test time, the annotated anchor sentence is used. Anchor prediction is noisy (Ko et al., 2022). Since the overwhelming majority of the anchor sentence is the sentence preceding $E$ (Figure 3), we believe this has a limited effect on our conclusions while leading to better controlled experiments. We leave anchor prediction for future work.

|  | BERTScore | BLEU-4 |
|---|---|---|
| DCQA-base | 0.915 / 0.494 | 0.323 |
| DCQA-ft | 0.911 / 0.474 | 0.313 |
| INQ-GoldT-base | 0.901 / 0.414 | 0.253 |
| INQ-GoldT-ft | 0.908 / 0.453 | 0.295 |
| INQ-PredT | 0.902 / 0.421 | 0.260 |

Table 2: BERTScore (F / rescaled F) and BLEU-4 for generated questions.

The target prediction model was first trained on INQUISITIVE then fine-tuned on ELABQUD.[4] In the question generation model, we replace the gold target in INQ-GoldT-ft with the predicted target (for both training and testing), with the rest of the setup identical to INQ-GoldT-ft.

**Settings** We use the same train/validation/test splits as in Srikanth and Li (2021). All model input formats and hyperparameters are tabulated in the Appendix, Table 7.

### 4.2 Results

**Automatic evaluation** We first evaluate generated questions with two automatic measures, BERTScore (Zhang et al., 2019) and BLEU (Papineni et al., 2002), comparing the generated questions with human annotated questions.

For BERTScore, we include both the unscaled version and the rescaled version. The results are shown in Table 2. It is clear that our DCQA-based oracle models, exposed to the elaboration $E$, performs better than INQUISITIVE-based models. Fine-tuning with ELABQUD does not help with the oracle setup but improves substantially for INQUISITIVE-based models. INQ-PredT, which predicts the target span, shows a drop in performance in line with the observation in Ko et al. (2020), though still better than taking INQ-GoldT-base out-of-the-box.

**Human evaluation** We further perform human evaluation across three systems, taken from the stronger versions of each group: DCQA-base, INQ-GoldT-ft, and INQ-PredT. We evaluate questions with a framework adapted from the QUD human evaluation schema of Ko et al. (2023); annotators judge questions along two criteria:

(1) *Is the question reasonable to ask given the current context?* That is, is this a valid/reasonable

---

[4] Following the evaluation setup of (Ko et al., 2020): the span prediction model has a exact match of 48.05% and a precision of 83.6% on our test set.

| | | Reasonable? | Answered? |
|---|---|---|---|
| Human | Yes | 88 | 89 |
| | No | 12 | 11 |
| DCQA | Yes | 78 | 67 |
| | No | 22 | 33 |
| INQ-GoldT-ft | Yes | 42 | 18 |
| | No | 58 | 82 |
| INQ-Pred | Yes | 42 | 12 |
| | No | 58 | 88 |

Table 3: Human evaluation on generated questions; % of questions marked yes/no for each criterion.

| | BERTScore | BLEU |
|---|---|---|
| Context-only | 0.886 / 0.322 | 0.200 |
| Generic | 0.877 / 0.270 | 0.166 |
| Human question | 0.896 / 0.381 | 0.244 |
| DCQA-base | 0.894 / 0.374 | 0.248 |
| DCQA-ft | 0.891 / 0.353 | 0.226 |
| INQ-GoldT-base | 0.880 / 0.288 | 0.165 |
| INQ-GoldT-ft | 0.880 / 0.288 | 0.178 |
| INQ-PredT | 0.879 / 0.282 | 0.172 |

Table 4: BERTScore (F / rescaled F) and BLEU-4 for GPT-3 generated elaborations given different prompts.

QUD having read so far?

(2) *Is this question answered by the elaboration?* For both criteria, annotators mark "Yes" (allows minor spelling and grammar issues for (1)) or "No".

Two undergraduate annotators evaluated a random sample of 50 questions generated by these three models along with the human annotated questions, with a total of 200 questions.

They agree 70.0% of the time for criterion 1 and 79.5% of the time for criterion 2. Shown in Table 3, annotators found human questions of the highest quality along both criteria, followed by DCQA-base, then INQ-GoldT-ft, and finally INQ-PredT. This is in-line with the automatic evaluation results. Interestingly, annotators report that both INQUISITIVE models perform worse on criterion 2 than 1, indicating that some of these questions may be valid QUDs but do not match the direction of the human elaboration. Consider the following elaboration in context:

(2) Should kids play tackle football? Football is a rough game. **E**: Players get bounced around.

A QUD like *Why is football a rough game?* makes the most sense for the actual elaboration "Players get bounced around", but a question such as the one generated by INQ-GoldT-ft, *What happens to players who get hurt playing football?*, is not answered even though it is a valid QUD.

## 5 Zero-shot elaboration generation

Finally, we experiment with the utility of questions on elaboration generation, i.e., task (2) in Section 4: given $C$ and $Q$, generate elaboration $E$ by modeling $P(\mathbf{e}|C, Q)$. Our hypothesis is that a good QUD should be able to guide a strong language model towards generating a better elaboration than *without* such guidance, in the sense that the elaboration should be more on-topic, and more frequently an explanation rather than simply continuing a story.

### 5.1 Setup

We use GPT-3 (Brown et al., 2020) for this task due to its vast text generation and open-domain question-answering capability (Liang et al., 2022). Specifically, we use `text-davinci-002` (temperature 0, max # tokens 128) with the following conditions, giving context consisting up to 5 sentences prior to the elaboration (this context window follows the same setup as Srikanth and Li 2021):

(1) **Context only**: the prompt consists of only the context;

(2) **Generic**: we concatenate the context with "*Please explain the last sentence in simple terms:*" as a generic instruction;

(3) **QUD**: we formulate the input as *Context: <context>, Question: <question>, Answer:*. We use questions both annotated questions from ELABQUD and ones from our question generation systems.

### 5.2 Results

**Automatic evaluation** We first report automatic evaluation results in Table 4, using BERTScore and BLEU with the gold elaboration as reference. The best performance is achieved with a QUD setup, a positive indication that our hypothesis holds: QUDs inform elaboration generation. However, this is more true for *good* QUD's generated by annotators, or by the DCQA models that were exposed to the elaboration itself. This result echoes what we found in Section 4.2: the INQUISITIVE models, although they often generate good QUDs, may not necessarily generate the QUD for the elaboration that the professional editor had in mind (as both BERTScore and BLEU compares the hypothesis with a reference). These cases lead to worse automatic measures compared to context-only settings, indicating a challenging future direction for QUD generation. Interestingly, we also note that

| | **Context**: "That's when I learned I was never going to write a real novel because I could not think that way." The Nobel Pirze is 112 years old. Munro is the 13th female literature winner. Munro grew up in Wingham, a traditional Canadian town west of Toronto. She tended to write about her experience there and her life after the social revolution of the 1960s. **Elaboration**: It was a time when people began to break free of traditional roles. |
| --- | --- |

| |
| --- |
| **Context only**: "I'm not a very good person to ask about the state of the novel," she said. |

| |
| --- |
| **Generic**: The last sentence is saying that Munro usually writes about her life in the small town she grew up in, and her life after the social revolution of the 1960s. |

| |
| --- |
| **Human question** *What happened during the social revolution of the 1960s?*: 
 The social revolution of the 1960s was a time of great change and upheaval. |

| |
| --- |
| **DCQA-base** *What type of social revolution was she talking about?*: 
 The social revolution of the 1960s was a time of great change in many Western societies. |

| |
| --- |
| **INQ-GoldT-ft** *What happened after the social revolution of the 1960s?*: 
 Munro continued to write about her experiences in Wingham and her life after the social revolution of the 1960s. |

| |
| --- |
| **INQ-PredT** *Who caused the changes in the community?*: 
 It is not clear who caused the changes in the community that Munro writes about. |

Table 5: An example of GPT-3 generated elaborations.

using a generic instruction does not yield better results than instead providing *no instruction* and only the context itself.

**Manual evaluation** We additionally perform human evaluation on the generated elaborations across these different prompts. In this setup, we mimic how elaborations would happen *organically* in a simplified document: a reader would *not* have access to the QUD but only to the generated elaboration, directly continuing prior discourse. A human evaluation would also reveal whether models generate elaborations that do not follow the exact direction from the document but are nonetheless good and plausible elaborations, an aspect that is not captured by the automatic measures.

Specifically, we provide two linguistics student annotators with a randomly sampled subset of 50 instances from the test set. The annotators were shown up to 5 sentences of prior context, then elaborations from GPT-3 as well as the original human elaboration from the document. These elaborations are randomly ordered. The annotators were asked to select and rank the top 2 elaborations independently along two criteria: (1) coherent continuation of discourse; (2) elaboration-like or explanation-like, rather than providing new information for story continuation (Srikanth and Li, 2021).

Table 6 shows that QUD-prompts produce more informative and on-topic elaborations, and so are ranked as highly elaboration-like. Take the context-only generation in Table 5; while it matches in style and is very fluent with the text (a very reasonable next line and quote from Munro), it is completely off-topic from the true elaboration, which describes

| | Elaboration-like | | Coherence | |
| --- | --- | --- | --- | --- |
| | #1 | #2 | #1 | #2 |
| Gold | 27 | 31 | 28 | 31 |
| Context-only | 8 | 16 | 10 | 14 |
| Generic | 1 | 3 | 2 | 3 |
| Human question | **19** | **19** | 21 | 14 |
| DCQA-base | 16 | 16 | **22** | **21** |
| INQ-GoldT-ft | **19** | 7 | 9 | 7 |
| INQ-PredT | 10 | 8 | 8 | 10 |

Table 6: Human evaluation of generated elaborations. % of times each system output is ranked #1 or #2 based on how elaboration-like and coherent the generation is, independently.

"the social revolution of the 1960s". Encouragingly, elaborations generated by human questions (and DCQA models) are ranked 1st most frequently (after the gold elaborations) in both criteria; this establishes the utility of good QUDs. For the INQ-style models, we see a clearer degradation in coherence despite them scoring well on Elaboration-like. We find that an off-topic question, like the one produced by INQ-PredT in Table 5, can easily throw off GPT-3. Generally, the generic-prompt and context-only elaborations are not similar to the human elaborations unless it is a description or definition would obviously come next. As such, the elaborations generated *without* QUDs cannot replicate more sophisticated human elaborations, where those generated with QUDs can.

## 6 Conclusion

This paper tackles the task of generating elaborations during text simplifcation. We adopt the

view of the Questions Under Discussion (QUD) discourse framework, and model elaboration as answers to implicit QUDs. We present ELABQUD, a dataset of annotated QUDs for elaborative simplification. We experiment with a question generation → elaboration generation pipeline. Results show that good QUDs provide valuable cues for zero-shot elaboration generation.

## 7 Limitations

This paper focuses on *how* to generate elaborations, rather than *when* to do so. Namely, we assume that we know which sentences constitute elaborations using Srikanth and Li (2021)'s dataset. We leave the *when* question to future work, noting that sentence-level elaboration is infrequent among the articles analyzed by Srikanth and Li (2021). At the same time, what constitutes difficult content is subjective or reader-specific. Future work can explore using QUD for elaborative simplification in an interactive manner. Additionally, the space of possible QUDs given context is large, posing challenges to INQUISITIVE-based systems for future work.

Another challenge with generating elaborations is inherit to elaborations themselves: because they contain information not present in the original text, they are hallucinations. It will be important to analyze the difference between helpful elaborations and undesirable hallucinations, and we leave this to future work.

We also note that we focused here on English text of a particular genre (news), and that results may not generalize to other languages or other genres of text.

Finally, we acknowledge that the landscape of LLMs are rapidly changing every few months, and we have not experimented with the strongest models (i.e., GPT-4). However, the space of possible elaborations prevents unconstrained generation; the utility of QUD is exactly to point to *what is elaborated*. As shown with our results, both question generation and elaboration generation benefit from stronger language models, hence we are optimistic about what stronger LLMs will bring to elaborative simplification with QUDs.

## Acknowledgements

This research was partially supported by National Science Foundation (NSF) grant IIS-2145479. We acknowledge the Texas Advanced Computing Center (TACC)[5] at UT Austin for many of the results within this paper. We also acknowledge Kathryn Kazanas and Keziah Kaylyn Reina for their help with the manual evaluation of generated questions and elaborations.

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

## A  Analysis of discourse relations

While QUDs provide fine-grained information about the goal of each elaboration, we complement this view by examining the discourse relations between an elaboration and its prior context $R_{\mathrm{pre}}(S_{i-1}, E)$. We use the relation taxonomy from the Penn Discourse Treebank (Prasad et al., 2008; Webber et al., 2019), a structural-neural framework that lays out the discourse relations between two text spans (i.e., arguments) including temporal, comparison, cause, etc.

Since most of the elaborations are intersentential implicit discourse relations that are still challenging for models to identify automatically (Atwell et al., 2021), we randomly sampled 51 elaborations for two expert linguists to annotate using the PDTB-3 (Webber et al., 2019) level-2 taxonomy. The two experts agreed for 40 of those, which we use in this analysis.[6]

Figure 4 shows the distribution of $R_{\mathrm{pre}}$, with PDTB-3 distributions for reference. Compared to PDTB-3, whose distribution came from news text, we observe many more *Expansion.Manner* relations associated with elaborations that explain the manner in which a situation in the pre-elaboration sentence was done. As expected, *Contingency.Cause* frequently appears. Our manual examination indicates that authors often stated the result in the complex explicitly and left cause implicit; when simplifying, this implicit cause was deemed too confusing for younger readers and so was added as the elaboration. *Expansion.Conjunction* is often linked with definitions. In many cases, an *EntRel* (entity relationship only) or a *NoRel* (no relation) involve organizational sentences (c.f. Section 3.1 example (1)) that opens succeeding discourse. We noticed many more *Hypophora* relations compared to PDTB-3; these are questions posed by the editors simplifying the document that guides children for what comes next.

We also report the most frequent discourse relations associated with each of the top 5 most frequent question type:

*Concept Q*: EntRel, Expansion.Conjunction
*Example Q*: Expansion.Conjunction, Contingency.Cause
*Consequence Q*: EntRel, Expansion.Conjunction
*Cause Q*: Contingency.Cause, EntRel

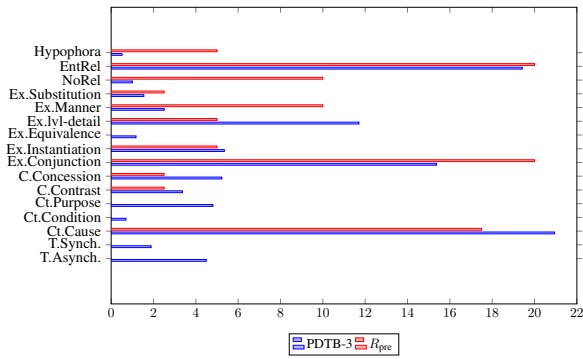

Figure 4: Relation distribution (%) in PDTB-3 and a sample of 40 agreed elaborations in ELABQUD.

*Procedural Q*: Expansion.Manner, Expansion.Conjunction

Overall, we observe a relatively high correlation between the type of questions and the discourse relations connecting an elaboration and its preceding context; both are informative in the type of content present in an elaboration.

## B  Model setup and hyperparameters

We tabulate all model setup and hyperparameters in Table 7.

## C  Copyrights

This work depends on the Newsela text simplification dataset (Xu et al., 2015). This dataset is free-to-use for academic researchers at https://newsela.com/data. The authors have obtained permission from Newsela to use this data.

## D  Compute

For all models in this work, we used 2 compute nodes each consisting of 3x NVIDIA A100 GPUs. All experiments finished in under 2 hours.

---

[6]A state-of-the-art classifier (Kim et al., 2020) did poorly on correctly classifying the relations with 42.5% accuracy on the 40 relations; thus, we do not include analyses from automatically recognized relations.

| Model | Input Format | Hyperparameters |
|---|---|---|
| DCQA-base | [context-dcqa],[anchor], [elaboration], [question] | learning_rate=5e-5, epochs=5,batch_size=8 |
| DCQA-ft | [context-dcqa], [anchor], [elaboration], [question] | learning_rate=2e-5, epochs=5, batch_size=2 |
| INQ-GoldT-base | [context-inq], [anchor w/ gold target], [question] | learning_rate=5e-5, epochs=7,batch_size=8 |
| INQ-GoldT-ft | [context-inq], [anchor w/ gold target], [, question] | learning_rate=2e-5, epochs=5, batch_size=2 |
| INQ-PredT | [context-inq], [anchor w/ predicted target], [question] | learning_rate=2e-5, epochs=5, batch_size=2 |
| Target prediction | [context], [anchor sentence], [gold span] | learning_rate=5e-5, epochs=3, batch_size=16 |

Table 7: Model settings. INQUISITIVE models (including the target prediction model) are reproduced from the same setup as Ko et al. (2020) before fine-tuning on ELABQUD. Models fine-tuned on ELABQUD is done with the same input format, where the hyperparameters denote training setup of the fine-tuning stage only. For DCQA models, *context-dcqa* denotes all sentences prior to the elaboration (where the anchor sentence is enclosed with a delimiter). For INQUISITIVE models, *context-inq* denotes all sentences prior to the anchor; the anchor includes the gold or predicted target denoted enclosed with a delimiter. For the target span prediction model, the SQuAD QA setup is followed as in Ko et al. (2020)'s span prediction model: SQuAD question → context, SQuAD context → anchor sentence, SQuAD answer → gold span. Questions are decoded with the HuggingFace default greedy decoding. All hyperparameters tuned on the validation set.