# OpenReview forum: "Elaborative Simplification as Implicit Questions Under Discussion"
_EMNLP/2023/Conference — EMNLP 2023 Main_

### Official Review · Reviewer_uFn6 · 2023-08-04

**Soundness:** 4

**Excitement:**

4: Strong: This paper deepens the understanding of some phenomenon or lowers the barriers to an existing research direction.

**Paper Topic And Main Contributions:**

Sequence to sequence models of simplification leave out of consideration the cases of elaborative simplification, where further information is added to the simplified text. Therefore, the authors make use of QUD theory to model elaborations as explicit answers to implicit questions.

The authors have created a corpus ElabQUD of elaborative simplifications that they enriched with respective QUDs.

This corpus they use to fine-tune a question generation model that they trained on two different QUD copora. These produced questions are then prompted to a GPT-3 model in order to produce real elaborations., which they claim to be of substantially higher quality.

Results of BLEU and BERTScore show that QUDs improve generation quality of elaborations after inputting the GPT-2 generated questions as prompts into GPT-3 for zero-shot learning. BUT this is true mainly for human-produced questions and the QUDs from models where the elaboration in the input was NOT masked!
INQUSITIVE models for which the elaboration was masked during training, did not produce QUDs that were useful in regard to the information requirements to which the elaboration is the answer. This leads to worse scores compared even to the context-only case.

**Questions For The Authors:**

I am accustomed to the circularity issue of QUD in NLG-related settings, so I immediately wondered why you actually trained a model for which the elaboration was not masked in the input. I mean, especially for generation, this is  important. One of the problems arising from this is the applicability of the model in a real text simplification scenario. Deciding when to generate an elaboration bases on the previous context, when the decision is made, you can only produce the QUD from previous context. Its therefore self-explanatory that the model without masked elaboration will perform best, but the performance that counts is the inquisitive type, which is reported to be worse than just feeding the context into GPT-3.
The question is: why do you conclude that its the QUD that improves the generation quality? Does it have to be a QUD? wouldn't be a representation that is not encoded in natural language be euqally usable and directly integratable in the generation (seq2seq) pipeline, as long as it encodes the info given in the QUD?

**Reasons To Accept:**

The paper sheds light on a topic that has not yet been thoroughly explored in NLP / NLG; namely QUD.
Although the paper is a bit euphemistic about the impact of QUD on generation improvement, I think the most
important findings of this paper are the things that do not correctly work. The authors give another prove about
the circularity issue of QUD in NLG-related tasks: The source text is needed to produce a QUD that truly captures
the anticipated informational piece and therefore serves the purpose of generating a good continuation. As the paper
shows, the quality is strongly reduced when the model that produces the QUD gets input where the source sentence
for the QUD is masked, which results in QUDs that do not fit human intuition about the informational requirement that
triggered the elaboration. That also happens for data-to-text generation, where QUDs actually lead off-topic, loss
of coherence by the LLM itself added on top. The finding the authors report are in alignment with my personal
experience of applying QUD theory in NLG, and I think these results are important and should being published.
As the authors themselves term it, "These cases lead to worse automatic measures compared to context-only set-
tings, indicating a challenging future direction for QUD generation."

**Reasons To Reject:**

There are some conceptual issues with the design of the setup from a QUD-theoretical perspective, namely what the questions under discussion are allowed to ask for. I personally do not mind those, but the theory-focused QUD community will.
There is a large discussion about topic, focus and non-at-issue in the QUD community, which I personally also dislike,
but the authors should mention what they interprete to be focus and topic structure in case of elaboration (or whether they even
treat this information as non-at-issue). Often enough, annotation guidelines (e.g. Riester et al.) destroy themselves when reaching
atomic information entities due to restrictions what the QUD may ask for. In case of atomic database queries, the distinction between
focus and topic starts to blur, resulting in paradox situations where QUDs may not ask for the only informational entity in the sentence they shall ask for in an NLG setting. This is a cumbersome discussion and of least importance from the perspective of NLG engineers (like myself), but the purely theoretical QUD  proponents will want to know details.

**Reproducibility:**

4: Could mostly reproduce the results, but there may be some variation because of sample variance or minor variations in their interpretation of the protocol or method.

**Reviewer Confidence:**

5: Positive that my evaluation is correct. I read the paper very carefully and I am very familiar with related work.

**Typos Grammar Style And Presentation Improvements:**

To reason though discourse p.2 line 158
simplifieddocument p.3 line 195
please reformulate in line 523/524

---

> ### Author Rebuttal · Authors · 2023-08-28
>
> Thank you for your feedback!
>
> _Re: focus and topic structure in QUD theory and  “the design of the setup from a QUD-theoretical perspective, namely what the questions under discussion are allowed to ask for”:_
>
> Thank you for bringing this up, we have not engaged with this in our paper but will add a discussion. We agree that it could be worthwhile to bring in some of the large literature on topic, focus and non-at-issue in the QUD literature. Our particular approach was fairly unconstrained in what QUD questions could ask, focusing on annotators providing natural intuition for their questions. We were clear on their task of providing questions which (1)  the elaboration answered and (2) were based on the prior context, as well as identifying a “target” (i.e. a sub-line selection of the prior text) that the elaboration is about (section 3.1 Annotation task, paragraphs 2 and 3, respectively). We found this satisfactory based on manual review of annotators (as described in section 3.1 Annotators), and ran a question-type classifier to understand what kinds of questions are being asked (section 3.2 What types of questions are  asked?, Table 1). While we acknowledge that not all theoretical perspectives would endorse this, we will clarify exactly what we see as the terms of QUD and what latent theoretical assumptions there are.
>
>  What could potentially be worth looking into is the connection between the “target” of those QUDs highlighted by the annotators and the information structure of the anchor sentence, but we have not analyzed whether the targets are more about the foci or the topics. We leave this to future work.
>
> _Re: why train a model where the elaboration is not masked:_
>
> We view this as some sort of silver “upper bound”. We certainly agree with the shortcomings of DCQA seeing the elaboration (section 4.1 Practical System); however, it still provides insight into the utility of QUDs in elaboration generation, as our manual analysis shows (Table 6). The DCQA model follows a “recovery” approach of QUD, and is complementary to the expectation-driven (Kehler and Rohde) or “potential questions” (Onea) approach as in Inquisitive questions. The question we were seeking to answer with DCQA models is: If there was a model that had access to what our human annotators had, what kind of performance are we looking at? So it is more of a feasibility test with the tools we have, and pointing out the gaps between this and Inquisitive-style questions, and giving us and the community a sense of the height of the hill to climb.
>
> _Re: “why do you conclude that it's the QUD that improves the generation quality? Does it have to be a QUD? wouldn't be a representation that is not encoded in natural language be equally usable and directly integratable in the generation (seq2seq) pipeline, as long as it encodes the info given in the QUD?”:_
>
> Our analysis shows that QUDs do help elaboration generation even when the elaboration is masked, as your manual evaluation of generated elaborations (Table 6) ranks INQ-GoldT generations as more elaboration-like than DCQA’s. Additionally, this analysis ranks all QUD-based approaches as more elaboration-like than the Context-Only approach. We agree that the information can be an encoded representation rather than an explicit QUD, however we believe that a question-answering format is more natural to today’s NLG systems when the amount of training data is not a lot. We believe there is much improvement to be made, in generating and evaluating both questions and elaborations, and we leave this to future work.

---

### Official Review · Reviewer_qEyA · 2023-08-06

**Soundness:** 4

**Excitement:**

4: Strong: This paper deepens the understanding of some phenomenon or lowers the barriers to an existing research direction.

**Paper Topic And Main Contributions:**

This article tackles a particularly difficult challenge in automatic text simplification (ATS), namely dealing with the implicit. It approaches by focusing on elaboration, i.e. simplifying by adding content, whereas traditional methods tend to act by deleting or reducing textual elements. Elaborative simplification makes sense in cases such as the need to explain a technical word (in specialized domains) or to explicit some implicit information in the text that a given reader might not be able to retrieve by itself. More precisely, the paper adopts the framework of Question Under Discussion (QUD), which postulates that each sentence in a text could be seen as the answer to an implicit or explicit question from prior context. The authors then introduce ELABQUD, an extension to the corpus collected by Srikanth and Li (2021). The initial corpus comprises 1299 elaborations, to which the authors have added 2878 possible questions leading to these elaborations. They also annotated the targets of each elaboration. This is the first contribution of the paper.

Based on this corpus, the authors investigated two research questions: (1) is it possible to automatically generate the question behind an elaboration (QUD)? and (2) does knowing this QUD help to generate elaboration in texts to be simplified? They first show that QUD can be generated in a promising way, although there is still much room for improvement. Then, they demonstrate that having access to the QUD significantly helps LLMs to generate valuable and coherent elaborations.

In my opinion, this paper deals with a very specific issue, which does not have a direct impact on the performance of ATS models. Nevertheless, it is also one of the rare first steps towards taking the implicit into account in applications linked to reading difficulty. What's more, the article is well written and the experiments were carried out rigorously. In particular, the authors combined an automatic evaluation with a manual evaluation on each occasion, which enabled them to remain connected to the data. The pattern of results also seems very clear and consistent, which adds to the solidity of the article.

**Questions For The Authors:**

l.298-306: You report percentage agreement of 69.9%. Could you please elaborate a bit about the type of disagreement that you have noticed?

l.310: If I got it correctly, your dataset includes 1,299 elaborations, thus 1,299 targets. As they are composed of an average of 4.54 tokens, I do not understand how you reach the number of 13,153 tokens.

**Reasons To Accept:**

- Original topic, that although it is quite focus, could have wider repercussions over time.
- Convincing methodology, at least as regards the resource building (for which inter-rator agreements are reported) and the evaluations (that combines automatic and manual evaluations).
- There are two contributions of the paper: a resource and one scientific clarification about the impact of QUD when generating elaborations.

**Reasons To Reject:**

- The topic and contributions could be seen as too focused.

**Reproducibility:**

4: Could mostly reproduce the results, but there may be some variation because of sample variance or minor variations in their interpretation of the protocol or method.

**Reviewer Confidence:**

3: Pretty sure, but there's a chance I missed something. Although I have a good feel for this area in general, I did not carefully check the paper's details, e.g., the math, experimental design, or novelty.

**Typos Grammar Style And Presentation Improvements:**

l.093: corpus of multiply annotated → multiply is probably not the correct term here
l.100: We find authors elaborate→We find that authors elaborate
l.193: as section 3 describes the ELABQUD corpus, I did not find the title of the section very coherent with the content. It seems that experiments will follow, whereas it is a resource.
l.195: For a simplifieddocument → For a simplified document
l.513: givng → giving

---

> ### Author Rebuttal · Authors · 2023-08-28
>
> Thank you for your positive feedback!.
>
> _Re: “topic and contributions could be seen as too focused”_:
> We believe elaborative simplification to be very helpful in automatic text simplification, especially when simplifying for those who have limited knowledge of the domain and so are most in need of new information. Thus, we believe exploring elaborative simplification is useful and warrants detailed theoretical treatment (e.g., by using the QUD approach we take).
>
> _Regarding your questions_: often there are multiple plausible targets for an elaboration; this is the main source of disagreement in the location of the target (section 3.2). Hence, we kept all targets provided by the annotators, so there are actually 2878 targets (= number of questions in the dataset (end of section 3.1); this is why there are 13153 tokens. Finally, thank you also for catching these grammatical and spelling errors! We will be sure to fix them.

---

### Official Review · Reviewer_iTg6 · 2023-08-07

**Soundness:** 5

**Excitement:**

4: Strong: This paper deepens the understanding of some phenomenon or lowers the barriers to an existing research direction.

**Missing References:**

The paper shows a good knowledge of the domain, there is not important missing reference.

**Paper Topic And Main Contributions:**

This paper presents experiments in automatic text simplification (ATS), and more specifically on an understudied aspect of it: the addition of information that improves clarity for the reader. This process is called elaboration. The authors explore how to generate elaboration with a framework called QUD (for question under discussion). In this framework, an elaboration is considered as an answer to an implicit question. They base their work on a previously released corpus of elaborations. The authors report experiments for which they produced an annotated corpus, named ElabQUD. The corpus is one contribution of the paper. Another contribution is an analysis on the types of elaborations that are found in Newsela. The last contribution consists in experiments on question generation and elaboration generation, with several different protocols. The results show that the proposed approach is promising for generating elaborations for ATS.

**Questions For The Authors:**

- Will the dataset be available freely or will it also require granted access to Newsela?
- Did you have a look at the readability of the elaborations that were generated? For example in Table 5, the last example shows a cleft sentence, which is something we would want to avoid in a simple text. Is it common? Were regularities observed regarding the output of the models?

**Reasons To Accept:**

- Simplification through content addition is neglected in the ATS literature, while it has been shown to play an important part in actual text simplification for humans.
- The idea of exploring the QUD framework for elaborations in ATS as it is done here is a nice idea, which the authors soundly executed.
- The paper is dense, as everything is thoroughly explained and detailed, and at the same time it remains easy to follow.

**Reasons To Reject:**

I do not have strong reasons to reject, but there is one non trivial point I feel is lacking.

- As it is rightly pointed out in the introduction, hallucinations are a well-known risk with elaborations. It is also a frequent behaviour of LLMs such as GPT3 that is used here. I was surprised to not see this issue discussed at all in the body of the paper. I think this is an important lack, especially as the authors report that 34% of the elaborations are explanations of concepts, which may require information that can not be deduced from the text itself. While I would have expected/liked so see experiments and analysis/discussion around this issue, I can understand it may be out of scope for this paper and too demanding for a camera-ready version. It should at least be mentioned as a future work and/or in the limitations section.

**Reproducibility:**

4: Could mostly reproduce the results, but there may be some variation because of sample variance or minor variations in their interpretation of the protocol or method.

**Reviewer Confidence:**

5: Positive that my evaluation is correct. I read the paper very carefully and I am very familiar with related work.

**Typos Grammar Style And Presentation Improvements:**

l. 195 simplifieddocument, missing space

---

> ### Author Rebuttal · Authors · 2023-08-28
>
> Thank you for your positive feedback!
>
> _Re: factuality of elaboration content:_
> Our focus is to show that our QUD-based methods improve generated elaborations (Table 6), and we leave much of the (important!) nuance, such as hallucination from LLMs, to future work; we will make this explicit in our conclusions and limitations sections.
>
> _Re: readability of elaborations:_
> We did look at readability of the generated elaborations: this falls under the “elaboration-like” criterion of the manual analysis in Table 6. We note some generalizations for elaboration generation in section 5.2 Manual Evaluation, but did not find any frequently occurring structures such as clefts.
>
> _Re: data sharing:_
> We will publicly share all annotations contributed in ELABQUD; note that the Newsela dataset, while free for academic and research use, does require permission (Appendix C).

---

### Meta-Review · Area_Chair_Aihs · 2023-09-18

**Recommendation:** 5

**Metareview:**

Reasons to Accept:
(1) The paper addresses a neglected aspect of abstractive text summarization (ATS) by focusing on content addition, which is essential for actual text simplification.
(2) The exploration of the Question Under Discussion (QUD) framework for generating elaborations in ATS is an innovative and well-executed idea.
(3) The paper is well-structured and thoroughly explains its concepts, making it easy to follow.

Reasons to Reject:
(1) There are no strong reasons to reject the paper, but it could benefit from discussing the risk of hallucinations in elaborations, especially given that some elaborations involve explanations of concepts. While this might be a topic for future work or limitations, addressing it would add depth to the paper.

The authors have provided reasonable responses regarding hallucinations and discussions on focus and topic structure.

---

### Decision · Program_Chairs · 2023-10-07

**Decision:**

Accept-Main

**Comment:**

Reasons to Accept:
(1) The paper addresses a neglected aspect of abstractive text summarization (ATS) by focusing on content addition, which is essential for actual text simplification.
(2) The exploration of the Question Under Discussion (QUD) framework for generating elaborations in ATS is an innovative and well-executed idea.
(3) The paper is well-structured and thoroughly explains its concepts, making it easy to follow.

Reasons to Reject:
(1) There are no strong reasons to reject the paper, but it could benefit from discussing the risk of hallucinations in elaborations, especially given that some elaborations involve explanations of concepts. While this might be a topic for future work or limitations, addressing it would add depth to the paper.

The authors have provided reasonable responses regarding hallucinations and discussions on focus and topic structure.